# A FAN1 point mutation associated with accelerated Huntington's disease progression alters its PCNA-mediated assembly on DNA

Jonas Aretz [1,3], Gayathri Jeyasankar [1,3], Anna Salerno-Kochan [1], Maren Thomsen[1], Gabriel Thieulin-Pardo[1], Tasir Haque [2], Edith Monteagudo[2], Dan Felsenfeld[2], Michael Finley [2], Thomas F. Vogt[2], Julien Boudet [2] ✉ & Brinda C. Prasad [2] ✉

FAN1 is an endo- and exo-nuclease involved in DNA and interstrand crosslink repair. Genome-wide association studies of people with Huntington's disease revealed a strong association between the FAN1 R507H mutation and early disease onset, however the underlying mechanism(s) remains unclear. FAN1 has previously been implicated in modulating triplet repeat expansion in a PCNA dependent manner. To examine the role of PCNA on FAN1 activation, we solved the cryo-EM structures of a PCNA–FAN1–DNA complex. Our findings reveal that the FAN1 R507 residue directly interacts with PCNA D232. Biophysical interaction studies demonstrated that FAN1 enhances the binding affinity of PCNA for DNA, a synergistic effect disrupted in mutants carrying the R507H mutation. In contrast, PCNA does not affect the affinity of FAN1 for DNA but does modulate FAN1 activity upon ternary complex formation. The weakened and functionally altered FAN1 R507H–PCNA–DNA complex may partly impair the FAN1-mediated repair of CAG extrahelical extrusions, providing a potential explanation for the mutation's role in accelerating disease progression.

Huntington's disease (HD) is a triplet repeat disorder characterized by the expansion of a CAG tract in the huntingtin gene (*HTT*) that encodes an expanded polyglutamine motif. Further expansion of this *HTT*-CAG tract in the somatic cells throughout a person's life is necessary for disease progression[1]. CAG repeat length correlates with the age of motor dysfunction onset; inheriting a longer CAG repeat increases the likelihood of developing the disease at a younger age[2]. Genome-wide association studies in people with HD (PwHD) have identified several genes that encode proteins involved in DNA mismatch repair (MMR) and modify disease onset, including *MLH1, PMS1, PMS2, MSH3, LIG1,*

and *FAN1*[1]. Among these, a coding variant resulting in FAN1 R507H protein is strongly correlated with an earlier onset of motor dysfunction[1,3].

FAN1 (FANCD2-FANCI-associated nuclease 1) is a DNA structure-dependent nuclease ubiquitously expressed in the central nervous system and peripheral tissues that plays a role in resolving interstrand crosslinks at replication forks[4]. It possesses both 5′ flap endonuclease and 5′-3′ exonuclease activities, allowing it to process DNA structures that arise during replication stress and repair pathways. FAN1 comprises a DNA-binding SAF-A/B, Acinus and PIAS (SAP) domain, a

[1]Proteros biostructures GmbH, Bunsenstr. 7a, D - 82152 Martinsried, Germany. [2]CHDI Management, Inc, the company that manages the scientific activities of CHDI Foundation, Inc., Princeton, NJ 08540, USA. [3]These authors contributed equally: Jonas Aretz, Gayathri Jeyasankar. ✉e-mail: julien.boudet@chdifoundation.org; brinda.prasad@chdifoundation.org

tetratricopeptide repeat (TPR) domain that mediates dimerization and protein-protein interactions, a virus-type replication-repair nuclease module (VRR) containing the catalytic site, and an unstructured N-terminal region[5] (Supplementary Fig. 1F). This unstructured region contains several motifs that have been demonstrated to interact with PMS2, MLH1, FANCD2 and proliferating cell nuclear antigen (PCNA)[6–13].

Previous structural studies provided key insights into FAN1's mechanism of action, revealing its interactions with DNA and its unique "n + 3" exonucleolytic processing mode[14–16]. These studies revealed that FAN1 recognizes 5' flaps and branched DNA substrates through a sophisticated interaction network, demonstrating its role in DNA processing. Crystal structures of FAN1 bound to different DNA substrates elucidated how the catalytic domain engages DNA, identifying key residues involved in its nuclease activity[16].

Recent genetic studies have linked several coding mutations in FAN1 to repeat expansion disorders, particularly HD, including the FAN1 R507H mutation associated with the modulation of CAG repeat instability[6,7,17–20]. The damaging FAN1 R507H mutation is in a flexible loop within the SAP domain, not directly involved in DNA binding nor located within the catalytic center[21,22]. Accordingly, no impact of the R507H mutation on the DNA binding or nuclease activity of FAN1 has been documented to date[23]. The flexible loop of FAN1 plays a critical role in DNA-mediated dimerization, as evidenced by previous studies showing that deleting amino acids 510-518, or mutating residues K525E, R526E, and K528E, results in a dimerization-deficient mutant[14,24]. Subsequently, it was hypothesized that the R507H mutation might also disrupt FAN1 dimerization. However, the dimerization capability of the FAN1 R507H mutant has been demonstrated to be identical to that of wild-type protein[23], leaving the pathogenic mechanism and the structure-function relationship of the R507H mutation unresolved.

The role of PCNA has been highlighted as a potential activator of FAN1, particularly in the repair of short CAG-triplet repeats[6]. PCNA is a homotrimeric ring-shaped DNA clamp that encircles double-stranded DNA after being loaded by replication factor C (RFC)[25]. It serves as a processivity factor for DNA polymerase δ and is essential for both DNA replication and repair through a PCNA-interacting peptide (PIP) box[26]. However, the molecular basis of FAN1-PCNA interactions remains unclear.

Here, we present structural and biochemical insights into the regulation of FAN1 by PCNA and how the R507H mutation weakens this interaction. We report cryo-electron microscopy (cryo-EM) structures of both wild-type (WT) and R507H mutant FAN1 bound to PCNA and the canonical FAN1 DNA substrate. These high-resolution structures reveal that the R507H mutation disrupts a key salt bridge between FAN1 and PCNA, thereby weakening their interaction. This subtle alteration correlates with compromised PCNA-mediated modulation of FAN1 activity, as demonstrated by our in vitro nuclease assays. We further evaluate FAN1's nuclease activity on CAG repeat-containing DNA substrates representative of the presumptive DNA intermediates recognized by DNA repair pathways in repeat expansion disorders. By integrating structural, biochemical, and biophysical studies, our findings underscore how PCNA modulates FAN1's nuclease function and contribute insights that provide a mechanistic basis for the FAN1 R507H variant accelerating DNA repeat expansions. These results significantly contribute to our understanding of FAN1's regulation and its implications in genome stability and disease.

## Results

### FAN1 forms a ternary complex with 5'flap double stranded DNA and PCNA through R507 residue

To structurally evaluate the FAN1 R507H mutation, we initially attempted to crystallize FAN1 R507H in the presence of 5' flap double-stranded DNA substrates[16]. We produced and crystallized the R507H variant in different constructs derived from the N-terminally truncated FAN1 (trFAN1), which lacks the first 370 amino acids (Supplementary

Fig. 1). Although many variants successfully crystallized, only the crystals of the R507H variant bound to DNA, which also included the thermostabilizing K794A mutation and a deletion of the dimerization loop (Δ510-518)[21], diffracted to a high resolution of 2.65 Å (PDB: 8S5A, Supplementary Fig. 2). While H507 was resolved in the electron density, the proximity of the 510-518 loop deletion to the site of interest raised concerns about potential artefacts.

Hence, we used cryo-EM to investigate the R507H mutation, as this method does not require deletions or thermostabilizing mutations. First, we successfully solved the cryo-EM structure of the DNA-bound wild-type full-length FAN1 at 3.2 Å (PDB: 9EO1, Fig. 1B, Supplementary Fig. 3). For the structural studies, we chose the canonical double flap DNA substrate over a CAG extrahelical extrusion containing DNA, because the 5' phosphate of the post-nick strand is essential for tight coordination of the DNA to the NUC domain. This reduces the flexibility of this domain, resulting in more rigid particles and thereby enabling higher resolution structures. Indeed, the structural analysis revealed side-chain densities for the amino acids R706, H742, R952, R955, K986 in the NUC domain coordinating the 5'-phosphate. However, despite the high-resolution of the SAP domain, the flexibility of the dimerization loop prevented us from resolving R507 (Fig. 1D).

Previous biochemical studies[6,10] demonstrated that FAN1's interaction with PCNA is crucial for its localization to stalled replication forks, positioning it as a key regulator of FAN1's repair activity. Given these results, we aimed to further understand if PCNA could stabilize the FAN1 structure and provide an improved resolution of the R507 residue by solving the cryo-EM structure of full-length FAN1 in the presence of a 5' flap double-stranded DNA substrate and full-length PCNA.

We successfully obtained a cryo-EM density map of the wildtype FAN1–PCNA–DNA ternary complex at 3.27 Å resolution (PDB: 9EOA, Fig. 1A, Supplementary Fig. 4). This map revealed the active site of FAN1 within the NUC domain and, crucially, resolved the density for residues near the dimerization loop, including R507 (Fig. 1D). In contrast, this region remained unresolved in our previous structure of the FAN1–DNA complex, highlighting the critical role of PCNA in stabilizing FAN1. With this cryo-EM density map, we observed a direct interaction between R507 in FAN1 and D232 in PCNA (Fig. 1D). Additional structural comparisons between the FAN1–DNA and FAN1–PCNA–DNA complexes revealed that the angle of the 5' flap double-stranded DNA substrate remains fixed at 68° (Fig. 1E), and no significant conformational changes were observed in FAN1 between the two complexes (Fig. 1F). The detailed protein-DNA interactions are not accessible with our current dataset.

To elucidate the impact of the HD-associated R507H mutation in FAN1, we generated a full-length recombinant FAN1 protein bearing the R507H mutation and reconstituted the ternary complex with PCNA and 5' flap double-stranded DNA substrate. This yielded a high-resolution cryo-EM structure of the FAN1 R507H–PCNA–DNA complex at 3.47 Å (PDB: 9GY0, Fig. 1C, Supplementary Fig. 5). In this structure, the dimerization loop of FAN1 is largely resolved, extending towards the inter-domain connecting loop (IDCL), a well-established PIP-box interaction region in PCNA[27]. Although the overall conformation of FAN1 and the angle of the DNA substrate remained unchanged between the FAN1–PCNA–DNA and FAN1 R507H–PCNA–DNA complexes, notable conformational alterations were observed in PCNA (Figs. 1G, 2C). Specifically, the R507H mutation resulted in an increased distance of 16.5 Å between H507 of FAN1 and D232 of PCNA (Fig. 2A). This shift may allow for the observed interaction between H485 of FAN1 and Y211 of PCNA (Fig. 2B), a residue previously shown to play a critical role in modulating PCNA function[28].

To investigate the structural basis of FAN1-PCNA binding to (CAG)2 loop DNA, we present a cryo-EM analysis of the FAN1–PCNA–(CAG)2 loop DNA complex at 3.6 Å resolution (Supplementary Figs. 6, 7A–D). Although the flexibility of FAN1 bound to the (CAG)2 loop results in a less defined density of the complex compared to flap DNA, this

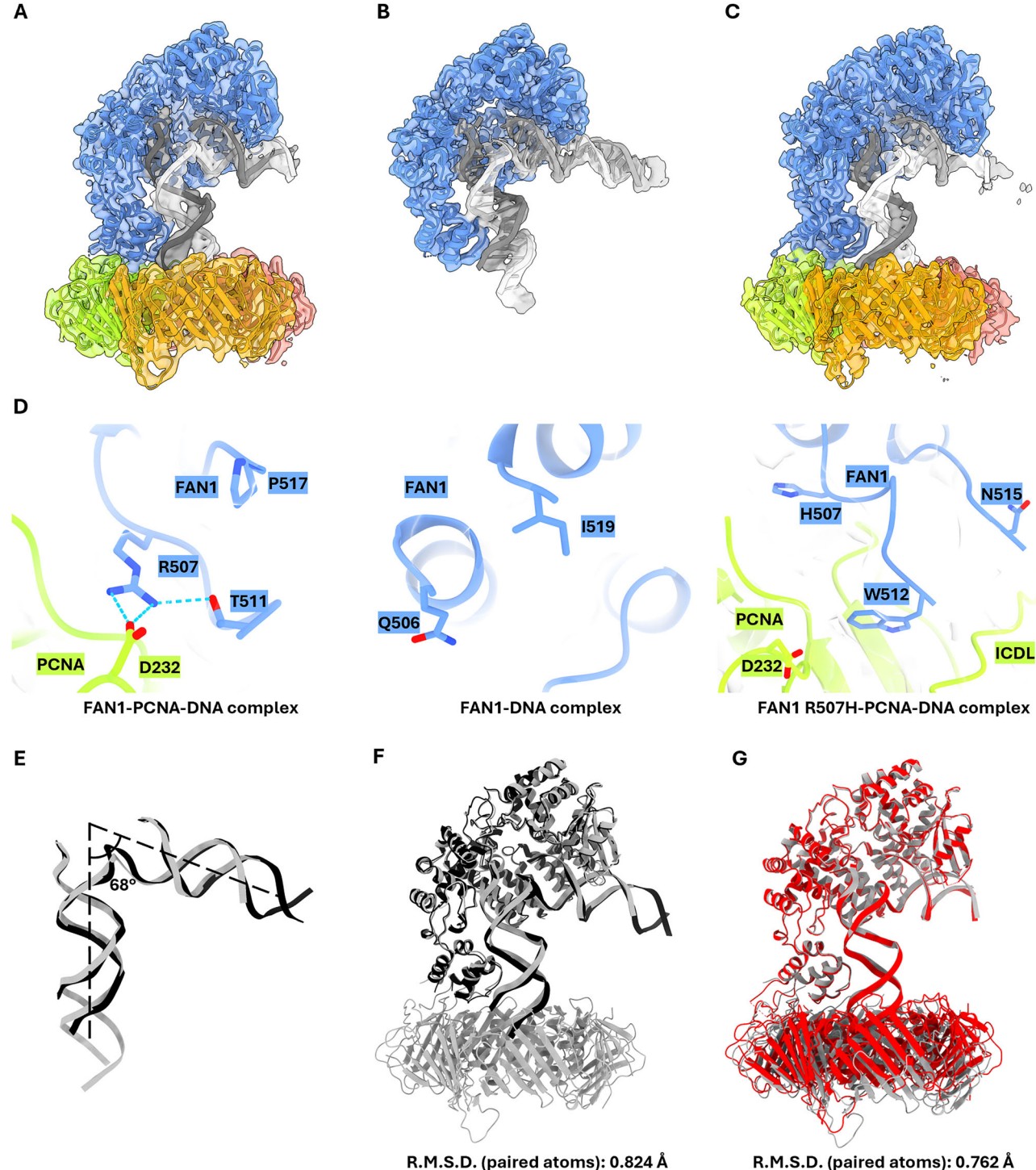

**Fig. 1 | Cryo-EM structural analysis of human FAN1, PCNA and 5′ flap double stranded DNA. A–C** Cryo-EM density maps fitted with their respective models of (**A**) the FAN1–PCNA–DNA ternary complex (9EO1) at 3.27 Å resolution, (**B**) the FAN1–DNA complex (9EOA) at 3.2 Å resolution, and (**C**) the FAN1 R507H–DNA complex (9GY0) at 3.42 Å resolution. The cryo-EM density maps are shown in surface as transparent density overlayed with a cartoon representation of the model. FAN1 is depicted in blue, PCNA in yellow/red/green, and DNA in gray/white. **D** Close-up views of key interactions. The salt bridge between R507 of FAN1 and D232 of PCNA stabilizes the FAN1-PCNA interface (left). The unresolved loop region of FAN1 in the FAN1–DNA complex (middle). In the mutant FAN1 R507H–PCNA–DNA complex, H507 of FAN1 remains distant from D232 of PCNA (right). **E** Superimposition of the DNA substrate models of the FAN1–PCNA–DNA (gray) and FAN1–DNA (blue) complexes. The angle of the DNA is at 68°. **F** Overall structural comparison between FAN1–PCNA–DNA (gray) and FAN1–DNA (black) complexes reveals no significant conformational changes in FAN1 upon binding to PCNA. **G** Overall structural comparison between FAN1–PCNA–DNA (gray) and FAN1 R507H–PCNA–DNA (red) complexes reveals conformational changes in the interaction of FAN1 R507H with PCNA.

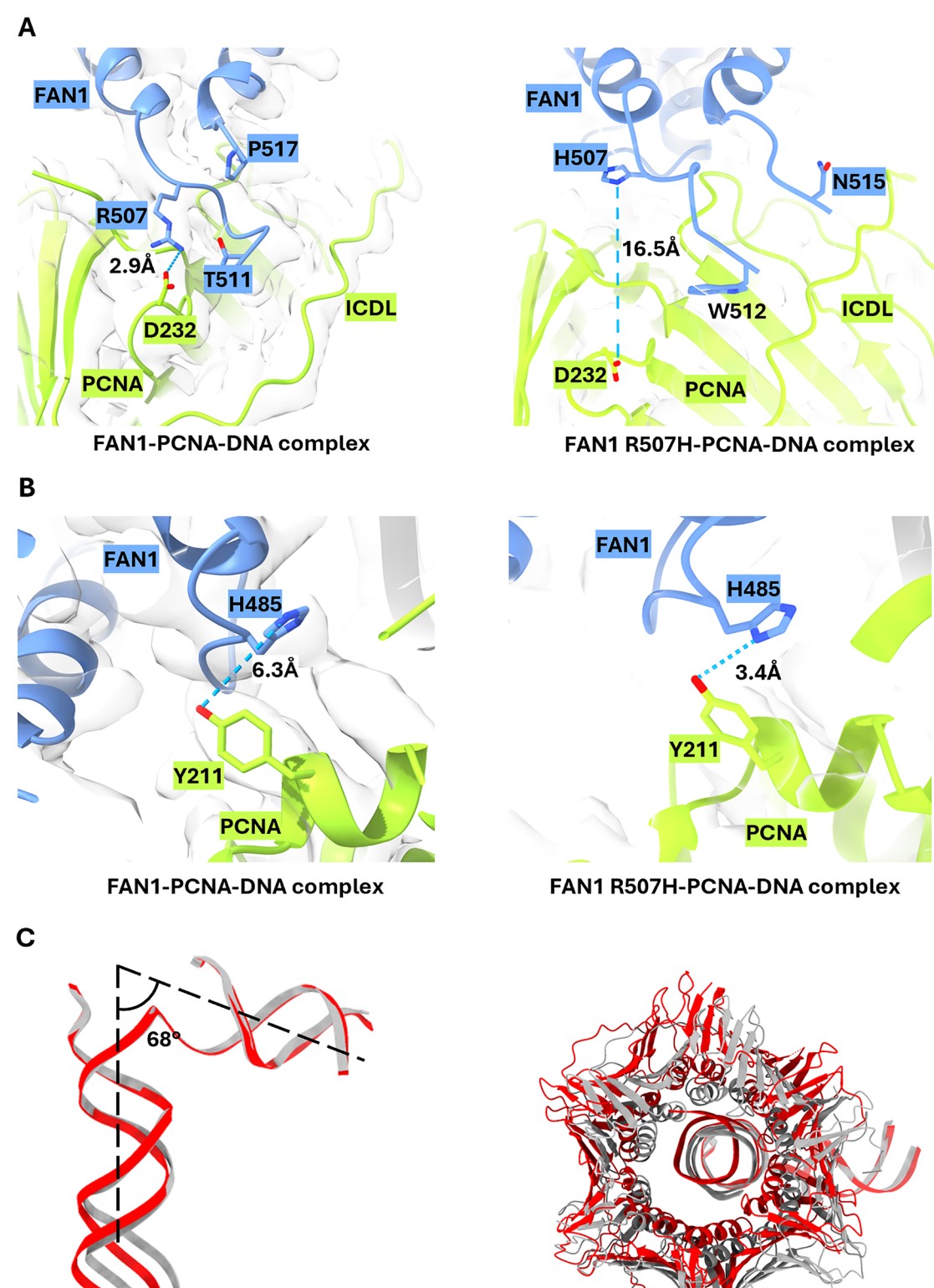

**Fig. 2 | Structural comparison of FAN1–PCNA–DNA with FAN1 R507H–PCNA–DNA complexes. A**, **B** Close-up views of key interactions and their distances. **A** The distance between R507 of FAN1 and D232 of PCNA in FAN1–PCNA–DNA complex is 2.9 Å (left) and in FAN1 R507H–PCNA–DNA complex is 16.5 Å (right). **B** The distance between H485 of FAN1 and Y211 of PCNA in FAN1–PCNA–DNA complex is 6.3 Å (left) and in FAN1 R507H–PCNA–DNA complex is 3.4 Å (right). **C** Superimposition of the DNA substrates of the FAN1–PCNA–DNA (gray) and FAN1 R507H–PCNA–DNA (red) complexes. The angle of the DNA is at 68° (left), and overall structural comparison between PCNA–DNA of FAN1–PCNA–DNA (gray) and FAN1 R507H–PCNA–DNA (red) complexes in orthogonal view showing the shift in PCNA orientation (right).

structure complements our findings that the FAN1-PCNA interaction interface is preserved across different DNA substrates.

## FAN1 R507 interacts with PCNA D232

To validate the identified interaction surface between FAN1 R507 and PCNA D232, we generated recombinant PCNA D232A and probed its ability to bind recombinant FAN1 R507H protein and form a ternary complex with DNA. For this purpose, we established a surface plasmon resonance (SPR) assay using FAN1 as a ligand and PCNA as an analyte. However, the proteins interacted with low affinity in the absence of DNA ($K_d > 30\,\mu M$, Supplementary Fig. 8A). Next, we immobilized DNA as ligand and probed its interaction with FAN1 and/ or PCNA as analyte (Fig. 3A, B). While PCNA was unable to bind DNA alone in the tested concentration range ($K_d > 10\,\mu M$; Supplementary Fig. 8B), in agreement with a reported NMR-derived $K_d$ of PCNA binding to DNA of 700 $\mu M$[29], its affinity was significantly increased in presence of FAN1 WT ($K_d = 1.6 \pm 0.9\,\mu M$) but to a lesser extent in presence of FAN1 R507H ($K_d = 3.0 \pm 0.8\,\mu M$; Fig. 3C). In addition to the FAN1 R507H, the affinity of PCNA for FAN1 WT-bound DNA was further decreased for PCNA D232A ($K_d = 3.7 \pm 1.7\,\mu M$) and for N-terminally truncated trFAN1 ($K_d = 3.9 \pm 1.3\,\mu M$; Fig. 3D). The lowest affinity was observed for PCNA D232A–trFAN1 R507H–DNA complex ($K_d = 14.0 \pm 5.1\,\mu M$) demonstrating that the efficient formation of a ternary complex between FAN1, PCNA and DNA requires at least two interactions with the FAN1 N-terminus and the support of the FAN1 R507 and PCNA D232 salt bridge. Interestingly, the decreased affinity upon disrupting the FAN1-PCNA interaction by mutation was mostly driven by a decreased on-rate (Fig. 3E), suggesting that in our biochemical system with short, open-ended DNA substrates, the FAN1–DNA complex is formed first and then recruits PCNA.

Compared to PCNA, FAN1 affinity for DNA was in the nanomolar range ($K_{d,FAN1\ WT} = 7.0 \pm 2.4\,nM$) and appeared to be unaffected by the presence of 10 $\mu M$ PCNA ($K_{d,FAN1\ WT\ +\ PCNA\ WT} = 7.2 \pm 2.4\,nM$; $K_{d,FAN1\ WT\ +\ PCNA\ D232A} = 5.4 \pm 1.7\,nM$; Supplementary Fig. 8D–G) in our experimental conditions. However, an increase in maximal response was observed upon FAN1–PCNA co-injection compared to FAN1 binding to DNA in absence of PCNA indicating successful ternary PCNA–FAN1–DNA complex formation; the maximal response in SPR experiments correlates with the molecular weight of the analyte (Supplementary Fig. 8G). To monitor the ternary complex formation in these co-injection experiments, we calculated the amplification of response by dividing the maximal response $R_{max}$ in the presence of PCNA by $R_{max}$ in the absence of PCNA from the steady state affinity fits (Supplementary Fig. 8H). While the amplification of response for co-injections of FAN1 WT and PCNA WT was $3.9 \pm 0.8$, this value was reduced for co-injections of FAN1 R507H and PCNA WT ($1.8 \pm 0.3$), FAN1 WT and PCNA D232A ($2.2 \pm 0.6$) and FAN1 R507H and PCNA D232A ($1.5 \pm 0.4$). These results further indicate that ternary PCNA–FAN1–DNA complex formation is impaired upon truncating the FAN1 N-terminus and introducing the FAN1 R507H or PCNA D232A mutations, which agrees with the PCNA titrations in presence of FAN1 (Fig. 3A–E).

Next, we validated these results with an orthogonal biophysical method by measuring binding of PCNA in presence and absence of FAN1 to Cy5-labeled DNA using microscale thermophoresis (MST; Fig. 3F; Supplementary Fig. 8I). In line with the SPR results, PCNA was unable to bind DNA alone ($K_d > 10\,\mu M$; Fig. 3G) and its affinity significantly increased in the presence of FAN1 WT ($K_d = 0.5 \pm 0.2\,\mu M$). This affinity increase was less pronounced for PCNA WT binding to DNA in the presence of FAN1 R507H ($K_d = 2.3 \pm 1.2\,\mu M$), as well as PCNA D232A binding to DNA in the presence of FAN1 WT ($K_d = 1.3 \pm 0.2\,\mu M$), FAN1 R507H ($K_d = 5.1 \pm 2.0\,\mu M$) or trFAN1 R507H ($K_d = 15.8 \pm 5.3\,\mu M$; Fig. 3H). Considering results from both biophysical methods to assess binding, FAN1 first binds DNA due to its higher affinity, followed by the recruitment of PCNA, which depends on stabilizing interactions

between PCNA and FAN1. However, in the cellular milieu, it is possible that RFC-loaded PCNA, already threaded on circular DNA may be playing a stabilizing role in the FAN1-DNA interaction. In summary, we posit that the PCNA-FAN1 interaction is mediated by the FAN1 N-terminus, confirming the previously proposed FAN1 interaction with PCNA via a non-canonical PCNA-Interacting Protein (PIP) motif[10], and the PCNA D232-FAN1 R507 interface as identified in the cryo-EM structure.

## Stability of the PCNA–FAN1–DNA complex is DNA substrate-dependent

Having established orthogonal binding assays, we tested the substrate specificity of the ternary PCNA–FAN1–DNA complex. First, we analyzed the affinity of PCNA for FAN1-bound heteroduplex DNA carrying one to four (CAG) extrahelical extrusions by MST and SPR (Fig. 4A, B; Supplementary Fig. 9A, B). While the affinity for all tested DNA substrates for either the N-terminal truncated FAN1, FAN1 R507H or PCNA D232A was reduced by nine- to forty-fold when measured by MST and SPR (Supplementary Fig. 9A, B), only minor differences of less than four-fold in affinity were detected for any (CAG) extrahelical extrusion containing DNA substrate compared to homoduplex DNA (Fig. 4A, B). Next, we determined the on- and off-rates from the SPR measurements (Fig. 4C, D; Supplementary Fig. 9C, D). The on-rates were mostly influenced by the FAN1-PCNA interaction leading to a four- to eight-fold decreased on-rate by introducing the FAN1 N-terminus truncation, FAN1 R507H and PCNA D232A (Fig. 4C; Supplementary Fig. 9C), and the off-rates were dependent on the DNA substrate showing the slowest off-rate on (CAG)2 DNA with $k_{off,(CAG)2} = 8.9 \pm 0.8\ 10^{-3}\,s^{-1}$ compared to homoduplex DNA with $k_{off,homoduplex} = 3.8 \pm 1.2\ 10^{-2}\,s^{-1}$ (Fig. 4D; Supplementary Fig. 9D). These results indicate that the assembly of the ternary PCNA–FAN1–DNA complex depends on the FAN1-PCNA interaction while its stability is determined by the DNA substrate.

## PCNA modulates FAN1 activity upon complex formation

Previously, PCNA was shown to modulate FAN1 endonuclease activity in biochemical assays, which could originate from a PCNA-induced change in DNA conformation or an allosteric activation of FAN1 by PCNA[6]. We hypothesized that PCNA-mediated FAN1 activity may be modulated by FAN1 mutations that impair ternary PCNA–FAN1–DNA complex formation.

To test this hypothesis, we developed an MST-based FAN1 endonuclease activity assay (Fig. 5A, B; Supplementary Fig. 10A–C) and assessed the ability of PCNA to influence FAN1 activity for Cy5-labeled 5′pG1/3′T1 dsDNA. The activity was quantified by Michaelis-Menten to determine Michaelis constant $K_m$ and the catalytic constant $k_{cat}$. Indeed, the addition of PCNA WT reduced activity of full length FAN1 below detection, independent of the R507H mutation. N-terminally truncated FAN1 WT, however, displayed a remaining low activity in presence of PCNA WT of $k_{cat} = 1.1 \pm 0.3\,s^{-1}$ compared to $k_{cat} = 4.2 \pm 1.2\,s^{-1}$ in absence of PCNA, and truncated FAN1 R507H was active in the presence ($k_{cat} = 5.8 \pm 1.5\,s^{-1}$) and absence of PCNA WT ($k_{cat} = 9.8 \pm 2.5\,s^{-1}$; Fig. 5C–E; Supplementary Fig. 10D–M). For PCNA D232A, trFAN1 WT ($k_{cat} = 5.5 \pm 2.4\,s^{-1}$) and R507H ($k_{cat} = 9.5 \pm 2.5\,s^{-1}$) activity remains unchanged while FAN1 WT and R507H activity could not and only partially ($k_{cat} = 2.0 \pm 0.7\,s^{-1}$) be restored, respectively (Fig. 5C–E; Supplementary Fig. 10M). Hence, the interaction with PCNA inhibited FAN1 activity on 5′ flap DNA substrates, which could be restored by truncating the FAN1 N-terminus and introducing the FAN1 R507H and PCNA D232A mutations. These observations agree with the SPR and MST affinity measurements that demonstrated that truncating the FAN1 N-terminus and mutating the FAN1 R507-PCNA D232 interface results in minimal FAN1–PCNA interaction. The $K_m$ values of FAN1 were identical in presence and absence of PCNA WT or D232A (Supplementary Fig. 10N–Q) for truncated FAN1 WT and

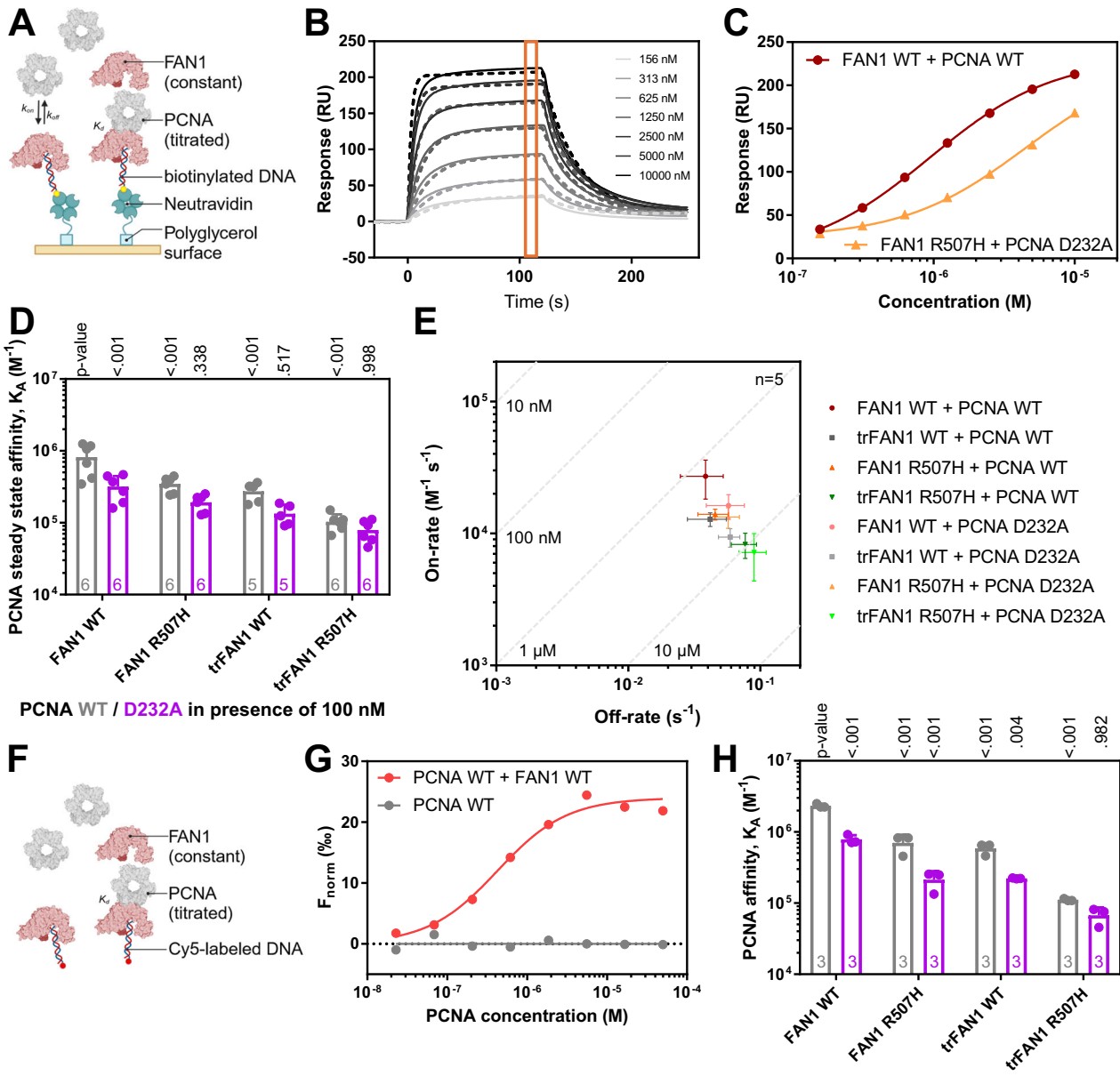

**Fig. 3 | FAN1 and PCNA form a ternary complex with DNA depending on the FAN1 N-terminus and the FAN1 R507-PCNA D232 interface. A** Complex formation of FAN1 and PCNA on DNA substrates was studied using SPR by titrating PCNA in the presence of 100 nM FAN1 using the A-B-A mode (Supplementary Fig. 7C). Created in BioRender[53]. **B** Exemplary sensorgrams of increasing PCNA concentrations binding to FAN1-bound 113 bp homoduplex DNA. The data (straight lines) was fitted with a 1:1 binding model (dashed lines). Responses for steady state affinity determination were read out 5 s before injection end (orange box). **C** Exemplary steady-state fit of PCNA WT binding to FAN1 WT-bound DNA (dark red circles) and PCNA D232A binding to FAN1 R507H-bound DNA (orange triangles). Data is fitted with a one-site binding model (lines). **D** SPR-derived steady-state affinities of PCNA WT (grey) and D232A (purple) for (tr)FAN1 WT and R507H-bound 113 bp homoduplex DNA. **E** On-off-rate chart for PCNA WT (dark) and PCNA D232A (light) interacting with FAN1 WT- (red circle), trFAN1 WT- (grey square), FAN1 R507H-

R507H. These results are in line with the SPR results showing that PCNA has no influence on FAN1–DNA affinity (Supplementary Fig. 10G).

Next, we tested dsDNA substrates with and without (CAG)1-4 extrahelical extrusions. As the FAN1 endonuclease activity for these substrates was strikingly lower compared to 5′pG1/3′T1 dsDNA, we

(orange triangle), and trFAN1 R507H-bound (green triangle) 113 bp homoduplex DNA. **F** Complex formation of FAN1 and PCNA on Cy5-labeled DNA substrates was studied in solution using MST by titrating PCNA in the presence of 100 nM FAN1. **G** Exemplary dose-response curve for PCNA WT binding to Cy5-labeled 61 bp homoduplex DNA in the presence (red) and absence (grey) of 100 nM FAN1 WT determined by MST. **H** MST-derived affinities of PCNA WT (grey) and D232A (purple) binding to Cy5-labeled 61 bp homoduplex DNA in the presence of 100 nM (tr)FAN1 WT and R507H. Bar charts represent the mean, and error bars the standard deviation of *n* independent measurements. *P* values are indicated above the bars comparing values to measurements in the presence of FAN1 WT for PCNA WT titrations using a two-way ANOVA with Dunnett's post hoc test, and measurements with PCNA D232A are compared to PCNA WT measurements using a two-way ANOVA with Sidak's post hoc test.

adjusted the activity assay by increasing the FAN1 concentration and the incubation temperature to 37 °C. However, this did not allow for the determination of reaction rates in time course experiments as FAN1 loses activity at higher temperatures after approximately 30 min. In the absence of PCNA, the MST signal of every tested DNA substrate changed slightly after incubation for 30 min in presence of FAN1, which

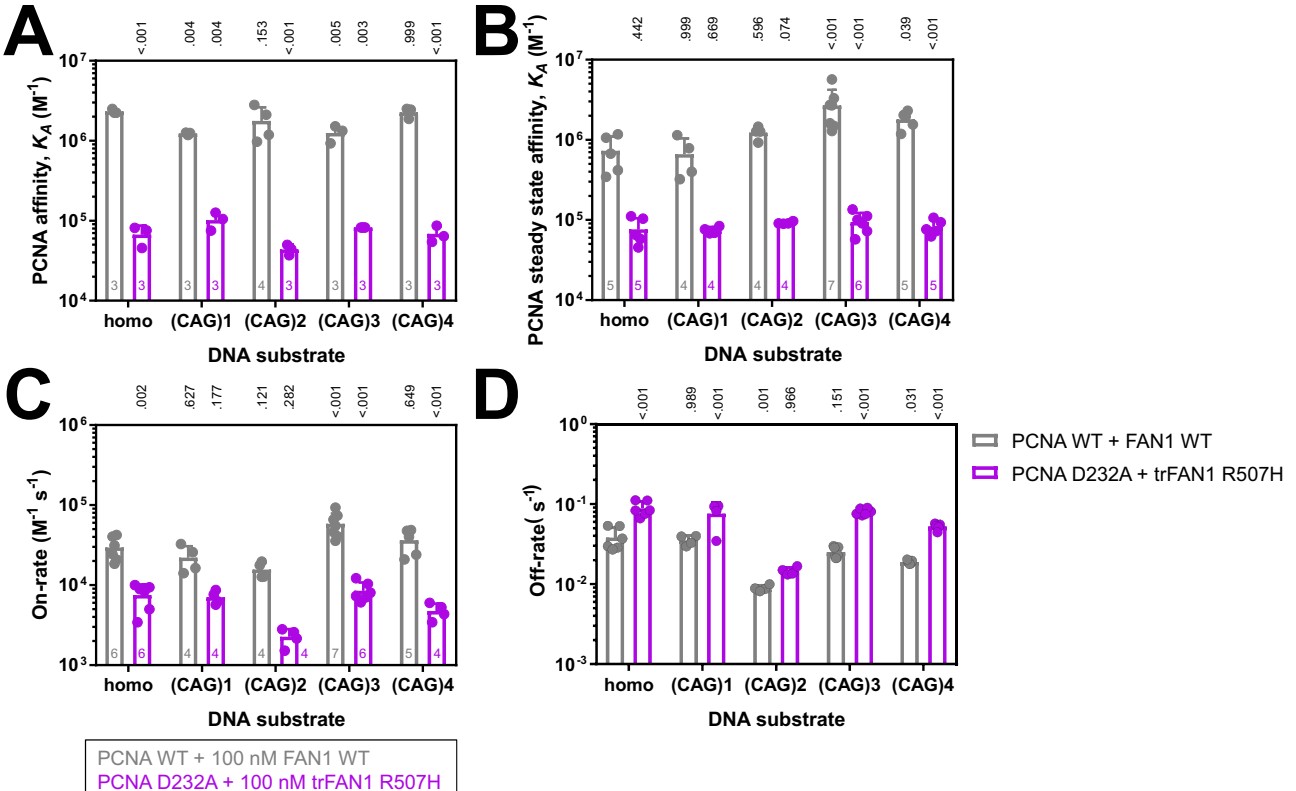

**Fig. 4 | Formation of the ternary complex between FAN1, PCNA and DNA determines specificity for (CAG) extrahelical extrusion containing dsDNA.**
**A** MST-derived affinities for PCNA WT binding in the presence of 100 nM FAN1 WT (grey) and PCNA D232A binding in the presence of 100 nM trFAN1 R507H to Cy5-labeled 61 bp homoduplex or (CAG)1, (CAG)2, (CAG)3 or (CAG)4 extrahelical extrusion containing dsDNA. **B** SPR-derived steady state affinities for PCNA WT binding in the presence of 100 nM FAN1 WT (grey) and PCNA D232A binding in the presence of 100 nM trFAN1 R507H to immobilized, biotinylated 113 bp homoduplex or (CAG)1, (CAG)2, (CAG)3, or (CAG)4 extrahelical extrusion containing dsDNA. **C**, **D** SPR-derived **C** on-rates and **D** off-rates for PCNA WT binding in the

presence of 100 nM FAN1 WT (grey) and PCNA D232A binding in presence of 100 nM trFAN1 R507H to immobilized, biotinylated 113 bp homoduplex or (CAG)1, (CAG)2, (CAG)3, or (CAG)4 extrahelical extrusion containing dsDNA. Bar charts represent the mean, and error bars the standard deviation of $n$ independent measurements. P values are indicated above the bars comparing values to homoduplex DNA for measurements with PCNA WT and FAN1 WT using a two-way ANOVA with Dunnett's post hoc test and measurements with PCNA D232A and trFAN1 R507H are compared to PCNA WT and FAN1 WT measurements for the individual DNA substrates using a two-way ANOVA with Sidak's post hoc test.

could be abrogated by introducing the catalytically dead D960A mutation pointing towards a low, basal endonuclease activity of FAN1 (Fig. 5F, G; Supplementary Fig. 11A–C). In the presence of PCNA WT, the largest change in MST signal was observed for (CAG)2 extrahelical extrusion containing dsDNA in the presence of FAN1 WT but not FAN1 D960A, indicating activation of FAN1 nuclease activity in the presence of PCNA (Fig. 5G, H). In contrast to the 5'pG1/3'T1 DNA substrate, PCNA-mediated activation of FAN1 WT was observed on dsDNA, which was higher compared to FAN1 with the N-terminal truncation, FAN1 R507H, and PCNA D232A (Fig. 5G). While no change in FAN1 activity upon PCNA addition was observed for homoduplex and (CAG)1 extrahelical extrusion containing dsDNA, PCNA activates FAN1 on (CAG)2, (CAG)3 and (CAG)4 extrahelical extrusion containing dsDNA (Fig. 5H; Supplementary Fig. 11A–C). Since the change in MST signal upon PCNA addition for the different DNA substrates correlated with the off-rates determined by SPR (Fig. 5I; Pearson correlation coefficient, p = 0.030), our results indicate that PCNA-mediated FAN1 activation depends on the stability of the ternary PCNA–FAN1–DNA complex and is most favored for dsDNA carrying a (CAG)2 extrahelical extrusion.

Taken together, our results demonstrate that FAN1 and PCNA form a ternary complex with DNA, that is required for PCNA-mediated FAN1 affinity modulation on different DNA substrates. This complex formation requires two interaction surfaces located in the FAN1 N-terminus and at R507 and is impaired by the disease-promoting R507H mutation.

## Discussion

To investigate the significance of the FAN1 R507H mutation, identified in human genetic studies as being correlated with an earlier onset of motor dysfunction[1,21], we initially used crystallography to determine the structure of the FAN1 R507H–DNA complex. The lack of certainty in the region of H507 arising from potential crystallization artefacts led us to employ cryo-EM for further analysis. We solved the cryo-EM structure of the full-length FAN1–DNA complex and found that the flexibility of the 510-519 loop hindered our ability to again resolve the R507 residue. Informed by studies demonstrating FAN1 activation by PCNA[6], we solved the cryo-EM structure of the ternary FAN1–PCNA–DNA complex. This structure revealed an interaction between PCNA residue D232 and FAN1 residue R507; the loop between residues 507-518, which was previously resolved in the FAN1 crystal structure (PDB 4RI8, chain A)[16], was not observed in our PCNA-free FAN1–DNA complex. However, in the presence of PCNA, we now observe this loop, suggesting that its stabilization is facilitated by the interaction between FAN1 R507 and PCNA. Furthermore, when the FAN1 R507 residue was mutated to H507, the interaction between PCNA D232 and FAN1 H507 was disrupted as evident by the increase in distance between these residues in FAN1 R507H–PCNA–DNA complex (Fig. 2A). These findings suggest that the mutation induces a conformational shift in FAN1, altering its structural alignment and enabling the formation of an interaction within the ternary complex with DNA.

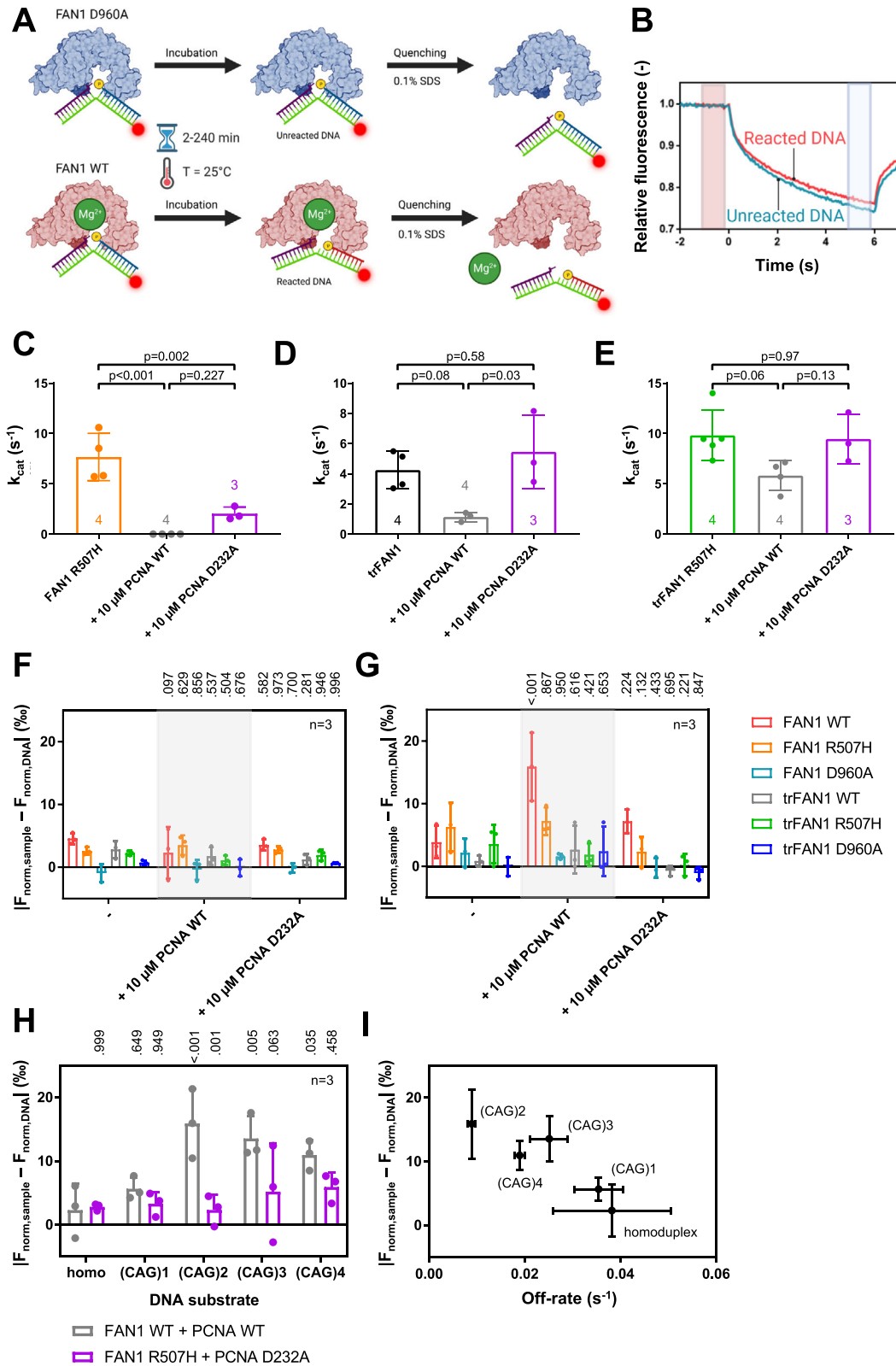

Furthermore, these observations indicate the FAN1 R507H mutation may involve an allosteric modulation of FAN1's activity. To corroborate our findings from the cryo-EM structure, we established biophysical binding assays to study the PCNA-FAN1 interaction. Interestingly, we could neither quantify the PCNA-FAN1 nor the PCNA-DNA interaction since these were in the high micromolar range. In contrast, the affinity of the interaction of PCNA with FAN1-bound DNA was observed to be

up to a thousand-fold higher, within the three-digit nanomolar range. The formation of the PCNA–FAN1–DNA ternary complex could be reduced but not abrogated by truncating the N-terminus of FAN1, which has been previously identified as a binding site for PCNA[10], or by introducing the disease-accelerating R507H mutation in FAN1 or the D232A mutation in PCNA, which was identified in our cryo-EM structure.

**Fig. 5 | Interaction of PCNA D232 with FAN1 R507 modulates FAN1 endonuclease activity. A** FAN1 was incubated with Cy5-labeled DNA stopping the reaction at different time points using SDS. By comparing samples incubated with FAN1 WT or the catalytically dead FAN1 D960A mutant. Created in BioRender. Aret, J. (2023) BioRender.com/d62t633. **B** reacted and unreacted DNA could be differentiated by their MST traces. **C–E** Catalytic efficiencies ($k_{cat}$) of **C** FAN1 R507H (orange), **D** trFAN1 WT (black), and **E** trFAN1 R507H (green) in presence and absence of PCNA WT (grey) or PCNA D232A (purple; p-values were calculated using a one-way ANOVA with Tukey test). **F, G** Changes in MST signal in presence of FAN1 WT (red), FAN1 R507H (orange), FAN1 D960A (light blue), trFAN1 WT (grey), trFAN1 R507H (green) and trFAN1 D960A (dark blue) compared to free DNA ($|F_{norm,sample} - F_{norm,DNA}|$) after 30 min incubation for **F** Cy5-labeled 61 bp homoduplex DNA and **G** Cy5-labeled 61 bp CAG2 extrahelical extrusion containing DNA in presence and absence of PCNA WT or D232A. P values are indicated above the bars comparing values in the presence of PCNA with measurements in the absence of PCNA using a two-way ANOVA with Dunnett's test. **H** Summary of changes in MST signal in the presence of FAN1 WT and PCNA WT (grey) or FAN1 R507H and PCNA D232A (purple) compared to free DNA ($|F_{norm,sample} - F_{norm,DNA}|$) after 30 min incubation for 1 nM Cy5-labeled 61 bp homoduplex or (CAG)1, (CAG)2, (CAG)3 or (CAG)4 extrahelical extrusion containing dsDNA. P values are indicated above the bars comparing values to homoduplex DNA for measurements with PCNA WT and FAN1 WT using a two-way ANOVA with Dunnett's test and measurements with PCNA D232A and trFAN1 R507H are compared to PCNA WT and FAN1 WT measurements for the individual DNA substrates using a two-way ANOVA with Sidak's test. **I** Correlation between MST-derived activity in the presence of FAN1 WT and PCNA WT (Fig. 5H) and SPR-derived off-rate for PCNA WT binding in the presence of FAN1 WT (Figs. 3, 4) for homoduplex or (CAG)1, (CAG)2, (CAG)3, or (CAG)4 extrahelical extrusion containing dsDNA substrates. Bar charts represent the mean, and error bars the standard deviation of $n$ independent measurements.

Our results indicate three findings. First, the FAN1–PCNA interaction surface is comprised of multiple weak interactions whose binding energies contribute to the overall binding affinity as demonstrated through biochemical analyses; the presence of DNA is also necessary for optimal binding, potentially facilitating the orientation of PCNA to interact with FAN1. Second, FAN1 recognizes DNA first due to its higher affinity for DNA, subsequently recruiting PCNA into a PCNA–FAN1–DNA complex in the described experimental conditions; this is corroborated by our kinetic measurements that demonstrate that the FAN1–PCNA interaction surface influences the on-rate. Third, the FAN1 R507H mutation results in a modest reduction in ternary complex formation with PCNA and DNA; it enabled us to solve the cryo-EM structure of the FAN1 R507H–PCNA–DNA complex. Structural comparisons between the FAN1–PCNA–DNA and FAN1 R507H–PCNA–DNA complexes reveal a slight shift in the alignment between PCNA and FAN1, suggesting a compensatory effect at the PCNA-FAN1 interface. This shift facilitates an interaction between H485 of FAN1 and Y211 of PCNA (Fig. 2B). These findings demonstrate that ternary complex formation is not completely disrupted by the R507H substitution in FAN1, suggesting that the PCNA–FAN1–DNA complex can still form in PwHD carrying this mutation.

To further address this key FAN1-PCNA interface with CAG repeat expansions, we resolved the cryo-EM structure of the FAN1–PCNA complex bound to a (CAG)2 loop DNA substrate at 3.6 Å resolution (Supplementary Figs. 6, 7A–C). This structure reveals that the FAN1-PCNA interface remains unchanged when compared to the previously resolved structure of FAN1–PCNA bound to a 5′ flap DNA substrate (PDB: 9EO1). Specifically, the critical interaction between FAN1 R507 and PCNA D232 is preserved. However, the flexible nature of the (CAG)2 loop DNA prevented us from achieving the same resolution as with the 5′ flap DNA substrate, likely contributing to the lower resolution of the FAN1 density in our study, which was also noted by Li et al. (2024)[30]. Furthermore, our structural findings are consistent with recent findings reporting a cryo-EM structure of FAN1 bound to (CAG)2 loop DNA and PCNA (PDB: 9CG4, 9CL7)[30]. When superimposed with our structure of FAN1–PCNA–5′ flap DNA (PDB: 9EO1), the overall architecture of the FAN1-PCNA interface remains unchanged (Supplementary Fig. 7D). The close agreement between these independent structural studies underscores the robustness of FAN1-PCNA-DNA interactions, regardless of the type of DNA substrate.

The PCNA-mediated FAN1 nuclease activity on DNA substrates with (CAG) extrahelical extrusions of varying sizes exhibited a correlation with the off-rate, which in turn correlates with the stability of the ternary PCNA–FAN1–DNA complex. A potential explanation for this observed correlation is an allosteric increase in FAN1 activity, whereas the longer residence time of the ternary PCNA–FAN1–DNA complex might be a consequence of the PCNA-mediated activation. Alternatively, assuming FAN1 exhibits low, substrate-independent endonuclease activity, the enhanced processing of (CAG) extrahelical extrusion containing DNA substrates by FAN1 may result from the prolonged residence time within the ternary complex encompassing PCNA and DNA. As no correlation between off-rate and activity was observed for FAN1 R507H in the presence of PCNA D232A (Supplementary Fig. 11D), it can be inferred that the off-rate is solely determined by FAN1's DNA specificity. Given that FAN1-DNA affinity remained unaltered by PCNA (Supplementary Figs. 8G, 9M–O), FAN1 must first recognize the (CAG) extrahelical extrusion independently, which may be accompanied by a conformational change of the DNA, before engaging with PCNA. Conversely, the increase in FAN1 activity on (CAG) extrahelical extrusions in the presence of PCNA WT but not PCNA D232A indicates an allosteric activation of FAN1 via the FAN1 R507-PCNA D232 interface. It is conceivable that this allosteric activation is initiated by PCNA influencing the conformation of the flexible loop in FAN1 upon forming a ternary complex with DNA, as observed in cryo-EM.

In mouse models of HD, knock-out (KO) of *Mlh1, Mlh3, Msh3* or *Msh6* led to decreased whereas KO of *Fan1* led to increased (CAG)-repeat instability[31–34]. A double KO of *Fan1* and *Mlh1* in an HD mouse model showed somatic (CAG)-repeat expansion comparable to WT[31], suggesting that two competing DNA repair pathways may exist. One pathway driven by MutSβ/MutL leads to increased somatic repeat expansions and an earlier onset of HD, while the other driven by FAN1 provides protection against somatic repeat expansions. Given that MutL is recruited by MutSβ, it was proposed that a competition between FAN1 and MutSβ for small extrahelical CAG extrusions, as observed in biochemical experiments[6], underlies the protective effect of FAN1. This hypothesis is consistent with reports indicating that overexpression of the nuclease-dead FAN1 D960A mutant, which can still bind DNA, can confer protection in certain cellular models of somatic-repeat expansion[17] and reports of non-coding mutants that increase FAN1 expression levels and protect against early motor dysfunction onset[22].

In addition to FAN1 expression levels, nuclease activity and DNA binding ability have been demonstrated to correlate negatively with somatic *HTT*-CAG expansions in HD. This correlation indicates that impairment of DNA binding and/or nuclease activity may result in an earlier age of motor dysfunction onset[21]. Whole genome association studies have revealed that the strongest correlation between earlier onset of HD and a R507H mutation in FAN1 is present[3]. It was therefore hypothesized that FAN1 R507H DNA binding or nuclease activity was diminished[22]; however, the FAN1 R507H mutant is active and exhibits no notable differences in DNA binding compared to FAN1 WT (Fig. 5)[23]. These results indicate a potential mechanism in which FAN1 DNA binding and/or nuclease activity is regulated by a downstream factor to prevent somatic-repeat expansion, which is proposed by us and others to be PCNA (Fig. 6)[6,30]. The discovery of an interaction interface between FAN1 R507 and PCNA D232, as revealed by cryo-EM, biophysical interaction, and enzymatic activity assays, indicates that the

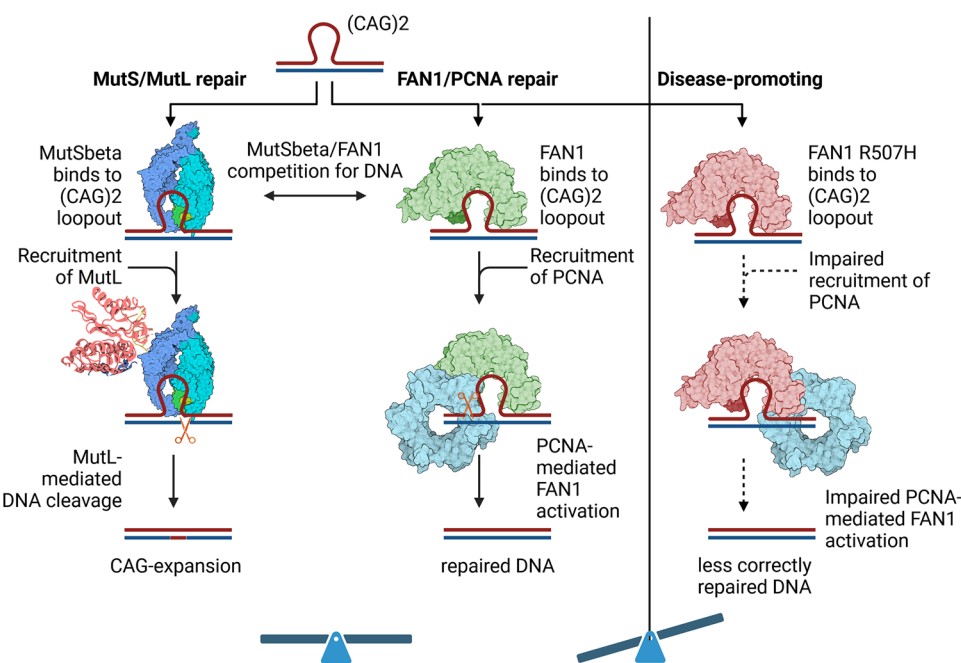

**Fig. 6 | Potential mechanism of FAN1-mediated repair of short (CAG)2 extra-helical extrusion containing dsDNA, which depends on PCNA-mediated activation of FAN1 nuclease activity.** The interaction with and consequently the activation by PCNA is impaired by the R507H mutation, leading to less efficient reduced FAN1-mediated DNA repair and more MutS/MutL-mediated DNA repair, which leads to CAG expansion. Created in BioRender. Aret, J. (2024) BioRender.com/z05q460.

reduced binding between PCNA and FAN1 R507H is a key factor in its disease-promoting effect. This results in a reduction in the efficiency of PCNA recruitment by FAN1 R507H in comparison to FAN1 WT, as evidenced by a reduced affinity of PCNA for FAN1 R507H and an impaired PCNA-mediated activation of FAN1 R507H nuclease activity for (CAG)2 extrahelical extrusions.

The protective effect of FAN1 against somatic-repeat expansion in HD raises a tantalizing role as a candidate therapeutic target. However, our proposed mechanism of FAN1 requiring PCNA-mediated activation to delay HD onset still needs to be validated in cell and animal studies. A first step could be the phenotypic comparison of FAN1 R507H with the PCNA D232A mutant identified here using knock-in cellular or HD mouse models. With these data, the development of allosteric FAN1 activators using the FAN1-PCNA interface could pave the way for future therapeutics for HD.

## Methods

### Plasmid generation
The coding sequences of the human FAN1 full-length, wild type, trFAN1 (364-1017), PCNA full-length, wild type, and mutated variants (FAN1 R507H, FAN1 D960A, PCNA D232A) were codon optimized for a suitable expression system and synthesized by Genscript (Genscript Biotech, New Jersey, USA). FAN1 and its variants were cloned with an N-terminal HIS-FLAG-MBP-tag and PreScission cleavage site. PCNA variants were cloned with N-terminal HIS-tag or HIS-AVI-tag followed by a PreScission cleavage site.

### Protein expression
FAN1 protein and its variants were expressed in the insect expression system. The recombinant bacmid and baculovirus stocks (P1, P2 & HTVS) were generated according to the manufacturer's instructions (Invitrogen by Life Technologies). The steps leading to the production of the virus stocks were performed using Sf9 cells and Sf-900™ II medium. The expression of the recombinant proteins was performed using Sf21 cells, with a final expression volume of 10 L of Sf-900™ II medium. The cell pellets were frozen at −80 °C until use.

PCNA and its mutated variant were expressed in *E. coli* BL21(DE3) strain induced with 0.5 mM IPTG and incubated for 18 h at 18 °C on a shaking platform. After the expression, cells were harvested and stored at −80 °C.

### Protein purification of human FAN1 protein and its variants
Pellets from baculovirus-infected Sf9 cells expressing the human FAN1 variants were resuspended on ice in buffer A (50 mM Tris-HCl, pH 8.0, 300 mM NaCl, 10 mM imidazole, 5% glycerol, 0.5 mM TCEP) supplemented with EDTA-free protease inhibitors. Cell lysis was achieved by Turrax homogenization on ice. The lysate was centrifuged at 75,000 *g* for 35 minutes at 10 °C. The clarified lysate was loaded onto a HisTrap column (2 × 5 mL) pre-equilibrated with buffer A. Elution was performed by a gradient (0-100% over 10 column volumes) of buffer B (50 mM Tris-HCl pH 8.0, 300 mM NaCl, 500 mM imidazole, 5% glycerol, 0.5 mM TCEP). The pooled fractions were treated by PreScission protease (in a 1:10 protease: target) for 16 h at 4 °C and dialyzed against buffer C (50 mM Tris-HCl, pH 7.4, 200 mM NaCl, 5% glycerol, 1 mM DTT). The sample was subsequently loaded onto a Heparin Sepharose column (2 × 5 mL) pre-equilibrated in buffer C supplemented with protease inhibitor cocktail tablets, EDTA-free. The bound proteins were eluted from the column by a gradient (0-100% over 20 column volumes) of buffer D (50 mM Tris-HCl, pH 7.4, 1 M NaCl, 5% glycerol, 1 mM DTT, protease inhibitor cocktail tablets). The fractions containing FAN1 were concentrated and loaded onto a S200 26/60 column pre-equilibrated in buffer E (20 mM Tris-HCl, pH 7.0, 200 mM NaCl, 5% Glycerol, 1 mM TCEP, protease inhibitor cocktail tablets). The pooled fractions from the size-exclusion chromatography were concentrated using an Amicon concentrator (MWCO 50 kDa) and snap-frozen in liquid nitrogen.

For cryo-EM applications, FAN1 variants were re-buffered to buffer F (20 mM Tris-HCl, pH 7.0, 200 mM NaCl, 1 mM TCEP). Previously purified protein was loaded onto a S200 10/300 GL column pre-equilibrated in buffer F supplemented with Protease Inhibitor Cocktail Tablets, EDTA-free. The fractions were analyzed by SDS-PAGE

and the fractions containing FAN1 were pooled and concentrated to 5 mg/ml.

### Protein purification of human PCNA protein and its variants

Pellets from *E. coli* cells expressing PCNA variants were resuspended in buffer G (1xPBS, pH 7.0, 1 mM DTT) supplemented with protease inhibitor cocktail tablets, EDTA-free, glycosidase, and DNAse on ice. Cell lysis was achieved by lysozyme and Sonifier ($4 \times 1$ min). The lysate was centrifuged at 75,000 g for 45 minutes at 10 °C. The supernatant (clarified lysate) was collected. The clarified lysate was loaded onto a HisTrap column (10 mL) pre-equilibrated in buffer H (1x PBS, pH 7.0, 137 mM NaCl, 5 mM imidazole, 1 mM DTT). The bound proteins were eluted by a gradient elution over 10 column volumes to 100% of buffer J (1x PBS, pH 7.0, 300 mM imidazole, 1 mM DTT). The pooled fractions were treated with PreScission protease (in a 1:20 protease: target) for 16 h at 4 °C and dialyzed against buffer G.

The cleaved target was loaded onto a HisTrap column (10 mL) pre-equilibrated with buffer H. Elution was performed by a step elution with 20% buffer J, followed by a step to 100% buffer J. The concentrated pool was loaded on an S200 26/60 column equilibrated in buffer K (20 mM Tris-HCl, pH 7.5, 150 mM NaCl, 10% glycerol, 2 mM DTT). The pooled fractions from the size-exclusion chromatography were concentrated and snap-frozen in liquid nitrogen.

For cryo-EM applications, PCNA variants were re-buffered to buffer F (20 mM Tris-HCl, pH 7.0, 200 mM NaCl, 1 mM TCEP). Previously purified protein was loaded onto a S200 10/300 GL column pre-equilibrated in buffer F supplemented with protease inhibitor cocktail tablets, EDTA-free. The fractions were analyzed by SDS-PAGE, and the fractions containing PCNA variants were pooled and concentrated to 5 mg/ml.

### Generation of dsDNA substrates

The single-stranded DNAs were synthesized by Metabion (Metabion International AG, Planegg, Germany), the sequences can be found in SI-Table 1. All DNA substrates were prepared following the same protocol. Equimolar amounts of complementary ssDNA strands were subjected to thermal denaturation at 95 °C for 10 minutes, followed by a 2h hybridization at room temperature (RT). Annealed dsDNAs were resolved on a 3–12% NativePAGE Bis-Tris Gel using a Tris-acetate-EDTA (TAE) buffer system. Electrophoresis was conducted at 180 V for 50 minutes at 4 °C. The resulting gel was stained with SYBR™ Safe DNA Gel Stain and visualized under UV light.

### Surface plasmon resonance (SPR)

**Immobilized DNA on sensor chip surface.** All measurements were performed on a Biacore 8k(+) (Cytiva) at 25 °C using a flow rate of 10 µL min⁻¹ for the immobilization and 30 µL min⁻¹ for kinetic measurements.

Neutravidin was used at a final concentration of 100 µg mL⁻¹ in 10 mM acetate buffer, pH 4.0, to be immobilized on the active and reference flow cell on carboxymethylated dendritic polyglycerol (CMPG, Xantec) after 30 s activation with NHS/EDC to about 3000 RU. Subsequently, biotinylated DNA was immobilized on the active flow cell of this neutravidin-coated surface at concentrations ranging from 3 to 25 nM to about 10 – 50 RU. The buffer for immobilization runs was 25 mM HEPES, pH 7.8, 150 mM sodium chloride, 0.05% Tween-20.

For the FAN1 titrations, FAN1 was diluted in eight concentrations ranging from 25 nM to 195 pM of FAN1 in the presence and absence of 10 µM PCNA in assay buffer (25 mM HEPES, pH 7.5, 125 mM KCl, 10 mM CaCl₂, 1 mM TCEP, 5% glycerol, 0.05% Tween-20, 0.05% BSA) and injected sequentially for 160 s onto both flow cells. The surface was flushed for 160 s dissociation time before being regenerated with 0.1% SDS for 30 s to remove bound FAN1 and PCNA from the surface, respectively.

For the PCNA titrations, the A-B-A mode was chosen. Solution A was 100 nM FAN1 in running buffer, whereas PCNA was titrated in eight concentrations ranging from 10 µM to 156 nM in the presence of 100 nM FAN1 as solution B. All solutions were applied with 120 s contact time before regenerating the surface with 0.1% SDS for 30 s.

**Immobilized FAN1 on sensor chip surface.** All measurements were performed on a Biacore 8k(+) (Cytiva) at 25 °C using a flow rate of 10 µL min⁻¹.

FAN1 was immobilized at a final concentration of 50 µg mL⁻¹ in 10 mM acetate buffer, pH 5.5, on the active flow cell of a carboxymethylated dendritic polyglycerol (CMPG, Xantec) chip after 120 s activation with NHS/EDC to about 250 RU. The buffer for immobilization runs was 25 mM HEPES, pH 7.8, 150 mM sodium chloride, 0.05% Tween-20.

For the kinetic measurements, DNA and PCNA were diluted in eight concentrations ranging from 1 µM to 0.46 nM and 50 µM to 3 nM, respectively, in assay buffer (25 mM HEPES, pH 7.5, 15 mM KCl, 10 mM CaCl₂, 1 mM TCEP, 5% glycerol, 0.05% Tween-20, 0.05% BSA) and injected sequentially for 120 s onto both flow cells. The surface was flushed for 240 s dissociation time before being regenerated with 1 M NaCl for 30 s to remove bound DNA or PCNA from the surface.

### Microscale thermophoresis (MST)

**PCNA binding assay.** PCNA was titrated in assay buffer (25 mM HEPES, pH 7.5, 125 mM KCl, 10 mM CaCl₂, 5% glycerol, 1 mM TCEP, 0.005% Tween-20, 0.1% BSA) before adding Cy5-labeled DNA to a final concentration of 10 nM in the presence or absence of 100 nM FAN1. The samples were incubated for 30 min at room temperature in the dark and centrifuged for 5 min.

All MST experiments were performed on a Nanotemper Monolith X or NT.115 pico using premium coated capillaries at 25 °C for 20 s at medium MST power in the PICO-RED acquisition mode recording a capillary scan before and after measurement to detect adsorption. The MST signal of the different time points was read out 1.5 s after turning on the infrared laser. Affinity was determined using the MO.AffinityAnalysis software (Nanotemper).

**FAN1 activity assay – 5'pG1/3'T1 dsDNA**. Unlabeled DNA was titrated in assay buffer (25 mM HEPES, pH 7.5, 15 mM KCl, 5 mM MgCl₂, 5% glycerol, 1 mM TCEP, 0.005% Tween-20, 0.1% BSA, and 3 µg mL⁻¹ plasmid DNA), before adding Cy5-labeled DNA dissolved in assay buffer with a final concentration of 1.25 nM. 135 µL of these stock solutions were transferred into 96 well plates and the reactions were started by adding 45 µL of a 10 nM FAN1 solution in assay buffer to obtain a final FAN1 concentration of 2.5 nM. The samples were shaken for 15 s at 1600 rpm to allow mixing and incubated at 25 °C in the dark. The reaction was stopped after 5, 10, 30, 60, 120, 180, and 240 min by transferring 20 µL from the reaction mix to 5 µL of 0.5% SDS solution.

All MST experiments were performed on a Nanotemper Monolith X or NT.115 pico using premium coated capillaries at 25 °C for 5 s at high MST power in the PICO-RED acquisition mode recording a capillary scan before and after measurement to detect adsorption. The MST signal of the different time points was read out 5 s after turning on the infrared laser, exported from the MO.AffinityAnalysis software (Nanotemper) and fitted to a one-phase association curve in GraphPad Prism 7.05 using the average $F_{norm}$ [‰] signal of unreacted DNA as baseline and the value of reacted DNA as plateau to obtain the reaction rate K [min⁻¹]:

$$Y = Baseline + (Plateau - Baseline)\left(1 - e^{-kx}\right) \qquad (1)$$

The reaction rate K [min⁻¹] was multiplied with the total DNA concentration (labeled + unlabeled DNA, [nmol]) in the sample to

obtain the initial reaction rate $v_0$ [nmol min$^{-1}$]

$$v_0 = k \cdot [DNA] \qquad (2)$$

that was plotted against the total DNA concentration [nmol] and fitted to a Michaelis-Menten model to obtain the limiting rate $v_{max}$ [nmol min$^{-1}$] and the $K_m$ value [nM]:

$$v_0 = v_{max} \frac{[DNA]}{K_m[DNA]} \qquad (3)$$

**FAN1 activity assay – 61 bp homoduplex and (CAG)1-4 extrahelical extrusion containing dsDNA.** 1 nM Cy5-labeled DNA was incubated in the presence and absence of 50 nM FAN1 in sample buffer (25 mM HEPES, pH 7.5, 90 mM KCl, 5 mM MgCl$_2$, 5% glycerol, 1 mM TCEP, 0.05% Tween-20, 0.1% BSA, and 3 µg mL$^{-1}$ plasmid DNA) in the presence and absence of 10 µM PCNA WT or D232A at 37 °C in the dark. The reaction was quenched after 30 min with 0.1% SDS.

All MST experiments were performed on a Nanotemper Monolith X or NT.115 pico using premium coated capillaries at 25 °C for 5 s at high MST power in the PICO-RED acquisition mode, recording a capillary scan before and after measurement to detect adsorption. The MST signal of the different time points was read out 5 s after turning on the infrared laser, exported from the MO.AffinityAnalysis software (Nanotemper) and responses were calculated by subtracting the responses in the presence of FAN1 ($F_{norm,sample}$) from the signals of DNA in the absence of FAN1 ($F_{norm,DNA}$):

$$Response = \left| F_{norm,sample} - F_{norm,DNA} \right| \qquad (4)$$

**Crystallization of FAN1**

The purified crystallization construct of FAN1(364-1017, R507H, K794A, Δ510-518) was mixed with purified 5′pG1/3′T4 DNA (sequence in Supplementary Table 1) in a 1:1.1 molar ratio and crystallized at a final protein concentration of 8 mg/ml. To find optimal crystallization conditions an initial screen was performed and hits in the Morpheus screen were further optimized in a grid screen. Best diffracting crystals were obtained at 20 °C through hanging drop vapor diffusion with 0.1 M of Morpheus Buffer 2 (Sodium HEPES; MOPS (acid)) pH 7.25, 0.1 M of Morpheus carboxylic acid mix (sodium formate; ammonium acetate; sodium citrate; sodium potassium tartrate (racemic); sodium oxamate), 32% (v/v) of Morpheus precipitant mix 2 (60% of each Ethylene glycol; PEG 8 K). Since these crystallisation conditions already had cryoprotectant properties, crystals were frozen directly in liquid nitrogen. The X-ray diffraction data from crystals of FAN1 with 5′pG1/3′T4 DNA have been collected at the CLS (Canadian Light Source Inc., Saskatoon, Canada) using cryogenic conditions. The crystals belong to space group P 2$_1$2$_1$2$_1$ with a unit cell of 91.6 Å 100.5 Å 112.9 Å 90° 90° 90° and one molecule in the asymmetric unit. Data were processed using autoPROC1.1.7 (XDS, POINTLESS 1 .12.14, AIMLESS 0.7.9, CCP48.0.004)[35–39]. The crystal structure was solved by molecular replacement using PHASER 2.8.3[40]. The published structure of FAN1 (PDB-ID 4RI8) was used as a search model. Final atomic coordinates were refined using Refmac5 and validated using MolProbity[41,42]. Structure analysis was performed in COOT, and figures were prepared using PyMOL[43,44]. For detailed data processing, refinement and validation statistics, please refer to Supplementary Table 2.

**Cryo-EM sample preparation**

Cryo-electron microscopy (cryo-EM) studies were performed on the full-length FAN1 protein in the presence of different dsDNA substrates (Supplementary Table 1) based on Wang et al.[16] and the presence of full-length PCNA. Concentrated samples of full-length FAN1 or full-length FAN1 R507H mutant and full-length PCNA were diluted to 3 µM in the buffer containing 20 mM Tris-HCl pH 7.0, 100 mM NaCl, and

1 mM TCEP. The dsDNA substrate was added to the diluted samples in an equimolar ratio, and the mixture was incubated in the presence of 1 mM CaCl$_2$ on ice for ~20–30 min. 4 µl of the mixture was then applied to QuantiFoil Cu/Au 300 M grids that have been glow discharged in a glow-discharger (Pelco). Grids were blotted (7–9 seconds) with filter papers pre-equilibrated for 45 minutes and then flash frozen in a liquid ethane-propane mixture using a Vitrobot Mk IV device (Thermo Fisher Scientific) set to 95% humidity and 4 °C.

**Cryo-EM data collection and processing**

All cryo-EM images were acquired on a Glacios transmission electron microscope (Thermo Fisher) operated at 200 keV in nanoprobe mode (C2 = 41.937%), equipped with a Falcon 4 electron detector at Proteros biostructures GmbH with a calibrated pixel size of 0.9142 Å/pix. All cryo-EM images were recorded using EPU.

Image processing was performed using cryoSPARC v4.3.0[45]. For FAN1–DNA complex, the dataset (5978 movie stacks) was imported in cryoSPARC followed by correction for drift and beam-induction motion and then used to determine defocus and other CTF-related values using CTFFIND. Only high-quality micrographs with low drift metrics, low astigmatism, and good agreement between experimental and calculated CTFs were further processed. Putative particles were automatically picked (~1.1 M) and subjected to reference-free 2D and subsequent 3D classification using C1 symmetry. Particles showing protein-like density features, especially features resembling secondary structure elements, were further refined using the non-uniform refinement strategy[46]. After per-particle motion correction and CTF refinement on 379,066 particles, the final reconstruction resulted in a refined cryo-EM density map of 3.20 Å (FSC criterion of 0.143). For FAN1–PCNA–DNA complex, the dataset (6289 movie stacks) was imported in cryoSPARC followed by correction for drift and beam-induction motion and then used to determine defocus and other CTF-related values using CTFFIND. Only high-quality micrographs with low drift metrics, low astigmatism, and good agreement between experimental and calculated CTFs were further processed. Putative particles were automatically picked (~2.3 M) and subjected to reference-free 2D and subsequent 3D classification using C1 symmetry. Particles showing protein-like density features, especially features resembling secondary structure elements, were further refined using the non-uniform refinement strategy. After per-particle motion correction and CTF refinement on 700,432 particles, the final reconstruction resulted in a refined cryo-EM density map of 3.27 Å (FSC criterion of 0.143). For FAN1 R507H–PCNA–DNA complex, the dataset (4664 movie stacks) was imported in cryoSPARC followed by correction for drift and beam-induction motion, and then used to determine defocus and other CTF-related values using CTFFIND. Only high-quality micrographs with low drift metrics, low astigmatism, and good agreement between experimental and calculated CTFs were further processed. Putative particles were automatically picked (~3 M) and subjected to reference-free 2D and subsequent 3D classification using C1 symmetry. Particles showing protein-like density features, especially features resembling secondary structure elements, were further refined using the non-uniform refinement strategy. After per-particle motion correction and CTF refinement on 487,606 particles, the final reconstruction resulted in a refined cryo-EM density map of 3.42 Å (FSC criterion of 0.143). For FAN1–PCNA–(CAG)2 loop DNA complex, the dataset (1458 movie stacks) was imported in cryoSPARC followed by correction for drift and beam-induction motion and then used to determine defocus and other CTF-related values using CTFFIND. Only high-quality micrographs with low drift metrics, low astigmatism, and good agreement between experimental and calculated CTFs were further processed. Putative particles were automatically picked (~1.2 M) and subjected to reference-free 2D and subsequent 3D classification using C1 symmetry. Particles showing protein-like density features, especially features resembling secondary structure elements, were further refined using the non-uniform refinement strategy. After per-particle motion

correction and CTF refinement on 165,802 particles, the final reconstruction resulted in a refined cryo-EM density map of 3.60 Å (FSC criterion of 0.143).

The local resolution for both structures was estimated using an FSC threshold of 0.5 and adaptive window factor of 6, which was then used for local filtering (Supplementary Figs. 3–6). DeepEMhancer[47] was used to post-process the half-maps using the highRes learning model to aid interpretation. Cryo-EM data collection parameters and processing statistics can be found in Supplementary Table 3 and the corresponding FSC to define resolution in Supplementary Figs. 3–6.

### Model building
To build an atomic structure of FAN1–DNA complex, the crystal structure of FAN1 (PDB: 8S5A) was docked into the refined 3D reconstruction map using UCSF (University of California at San Francisco) ChimeraX[48–50] and then manually rebuilt using COOT to fit the density. To build an atomic structure of FAN1–PCNA–DNA and FAN1 R507H–PCNA–DNA complex, the crystal structure of FAN1 (PDB: 8S5A) and the crystal structure of PCNA (PDB: 1VYM) was docked into the refined 3D reconstruction map using UCSF ChimeraX[48–50] and then manually rebuilt using COOT[43] to fit the density. The final atomic structure was refined in Phenix v1.21[51] and validated using MolProbity[52] and the half-map cross validation method (CCPEM package). Structure analysis was performed in COOT and ChimeraX, and figures were prepared using PyMOL and ChimeraX[44,48–50]. For refinement and validation statistics, please refer to Supplementary Table 3.

### Reporting summary
Further information on research design is available in the Nature Portfolio Reporting Summary linked to this article.

## Data availability
The crystal structure has been deposited in the Protein Data Bank (PDB) under the following accession codes: 8S5A (The crystal structure of FAN1 Nuclease bound to 5′ phosphorylated p(dG)/3′(dT-dT-dT- dT) double flap DNA). The cryo-EM structures have been deposited in the Electron Microscopy Data Bank (EMDB) and PDB under the following codes, respectively: EMD-19844 and 9EO1 (FAN1–DNA complex), EMD-19850 and 9EOA (FAN1–PCNA–DNA complex), EMD-51680 and 9GY0 (FAN1 R507H–PCNA–DNA complex), EMD-53279 (FAN1–PCNA–(CAG) 2 loop DNA complex). Source data files are published together with the manuscript.

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

## Acknowledgements
We thank A. Voelkl, T. Weinseis, H. Chepliaka, and S. Sharma for technical assistance in protein expression and purification, G. Zinner, I. Schleicher, and K. Pflügler for technical assistance with MST and SPR measurements, and J. Köhler for technical assistance with cryo-EM sample preparation. We thank Simon Noble for reviewing the manuscript. Cartoons in Figs. 3A, F, 5A, 6, and Supplementary Figs. 8C-E were created in BioRender. This work was supported by the nonprofit CHDI Foundation Inc. CHDI Foundation is a nonprofit biomedical research organization exclusively dedicated to collaboratively developing therapeutics that substantially improve the lives of those affected by Huntington's disease. CHDI Foundation conducts research in a number of different ways; for the purposes of this manuscript, all research was conceptualized, planned, and directed by all authors and conducted at Proteros Biostructures GmbH.

## Author contributions
M.T., B.C.P. and T.F.V. conceived and designed the study. G.T.-P. and A.S.-K. cloned and purified all described proteins and prepared DNA substrates. J.A. performed all biophysical and biochemical experiments in this study. M.T. crystallized and solved crystal structure. Model building and refinement was done by M.T. G.J. prepared cryo-EM samples, collected and processed cryo-EM data, built and analyzed all cryo-EM structures. J.A., G.J., A.S-K., M.T., T.H., E.M., D.F., M.F., J.B., B.C.P. were involved in interpreting the data. J.A., G.J., A.S-K., M.T., J.B., and B.C.P. wrote and revised the manuscript.

## Competing interests
The authors declare no competing interests.
