## [Transparent Peer Review file · Nature Communications]

A FAN1 point mutation associated with accelerated Huntington's disease progression alters its PCNA-mediated assembly on DNA

Corresponding Author: Dr Brinda Prasad

Version 0:

Reviewer comments:

Reviewer #1

(Remarks to the Author)

Review of the article " A FAN1 point mutation associated with accelerated Huntington's disease 2 progression alters its PCNA-mediated assembly on DNA " submitted by Aretz J et al

The manuscript by Aretz et al. makes a significant contribution to our understanding of FAN1 and PCNA's roles in Huntington's disease (HD), focusing on the R507H mutation associated with early-onset HD. The authors present a 2.6 Å X-ray structure of the FAN1 R507 mutant bound to DNA, alongside three cryo-EM structures of FAN1-DNA-PCNA complexes, including a ternary complex at 3.3 Å resolution. These structural insights illuminate the interactions within the ternary complex, revealing a direct interaction between R507 in FAN1 and D232 in PCNA. Notably, the DNA exhibits a 68° kink at the FAN1 active site.

Through SPR and MST analyses, the authors demonstrate that PCNA has weak intrinsic affinity for DNA, which is enhanced in the presence of FAN1. They also show that truncated FAN1 and mutations in R507 or D232 significantly reduce this affinity. Additionally, experiments using DNA with varying lengths of CAG repeats (1 to 4) showed no substantial differences in binding behavior.

The manuscript is well-written, with high-quality figures, and provides critical insights into DNA repair mechanisms and Huntington's disease pathology. This study represents an important advancement in the field and will appeal to a broad audience in molecular biology and neurodegenerative disease research.

Reviewer #2

(Remarks to the Author)

Reviewer #3

(Remarks to the Author)

FAN1 possesses both 5' flap endonuclease and 5'-3' exonuclease activities and is involved in repairing DNA inter-strand crosslink damage. The structure of FAN1 and how FAN1 recognizes the DNA 5' flap or a branched DNA (with both 3' and 5' flaps) were determined over 10 years ago (PMID: 25500724; PMID: 25430771; PMID: 25547363), along with the discovery of the "n + 3" exonuclease mechanism (PMID: 25430771). The current manuscript attempts to address the recent observation that FAN1 mutation R507H is associated with the CAG expansion in Huntington's disease patients, which compromises the PCNA-activated FAN1 activity on these CAG expansions. The authors report the structures of the WT and R507H mutant FAN1 bound to PCNA and a DNA substrate, leading to the discovery that the single mutation abolishes a salt bridge between FAN1 R507 and PCNA D232, thus weakening the interaction and explaining the compromised PCNA-stimulated FAN1 activity. They further performed in vitro binding and activity assays on DNA substrates containing 1 to 4

CAG repeat expansion, with overall anticipated results. A major weakness of the work is the use of flapped DNA substrate rather than a (CAG)_n bulge in their structural study, because FAN1 function on flapped DNA substrate is well understood thanks to previous structural and biochemical studies, and also because the DNA substrate used in structural work is different from biochemical assays which used CAG expansion DNA substrate, which is their current goal. Therefore, it remains unclear how FAN1 recognize and remove the CAG expansion. Overall, this work represents a modest incremental progress in this field, and one wonders if Communications Biology is a better fit.

Major concerns:

1. Previous key discoveries on the structure and mechanism of FAN1 (PMID: 25500724; PMID: 25430771; PMID: 25547363) were not prominently cited and summarized in Introduction. Only one of them is cited in Introduction for mutation effect (line 53); another one merely mentioned in Results for the selection of DNA substrates (line 74). It is rather disappointing, and such practice is not acceptable.

2. As mentioned above in summary, it is a major weakness of the work to use different DNA substrates for their structural and activity studies. The 5' flapped DNA substrate used in this work was already used in previous structural work. Therefore, the key question of how FAN1 recognizes CAG repeat expansion remains unclear.

2. Line 131 states in the presence of DNA, WT FAN1 and WT PCNA showed the highest binding affinity (Fig. 3C-D); but line 218 states "In line with our previous results, the PCNA-mediated activation of FAN1 WT was higher compared to FAN1 with the N-terminal truncation, FAN1 R507H and PCNA D232A (Fig. 5G)". And in Fig. 5C-D, with the presence of WT PCNA, the FAN1 R507, trFAN1 and trFAN1 R507 mutants had lower nuclease activity than without, and the first mutant had almost no activity. These observations are inconsistent.

3. Line 240, ... the previously unresolved loop between residues 507-518: Untrue, as this region was resolved in one of the crystallographic works mentioned above (PDB 4RI8 chain A).

Minor issue: Missing positive and negative controls in FAN1 endonuclease activity assays, i.e., with or without WT FAN1 in Fig. 5C-E.

Typos:

Line 15, two "Huntington's disease".

Line 26, add "_" to FAN1 R507H in "FAN1 R507H-PCNA-DNA", also replace the hyphen (-) with dash (–) as in line 22 for readability, also check other places, e.g., line 97.

Line 42-45, provide a domain architecture diagram of FAN1 can be helpful.

Line 71, use the prime symbol (5') rather than apostrophe (') throughout the text.

Line 83, Fig. 1B was cited before 1A.

Line 92, ... studies6..., cite more or use singular form.

Line 113, "ICDL" should be "IDCL".

Line 136, ... complex between FAN1, PCNA and DNA requires at least two interactions with the FAN1 N-terminus... the present data don't lead to such conclusion.

Line 201, "trFAN1 WT" is confusing, same for the figure legends e.g., Extended Data Fig. 8B. Also the number of kcat is inconsistent with Fig. 3D.

Line 416 and 729 these citations refer to the same paper.

Line 822, Fig. 1 should be Fig. 3.

Line 1016, missing concentration in Fig. 3B. The numbers above the histogram in Fig. 3D are confusing.

Line 1033, 1035, extra spaces within "acquired" and Local "Motion Correction" in Extended Data Figs. 3 and 4 legends.

Version 1:

Reviewer comments:

Reviewer #3

(Remarks to the Author)

The authors have address some of our concerns, but have not addressed an important concern - missing both positive and negative controls in FAN1 endonuclease activity assays, i.e., with or without WT FAN1 in Fig. 5C-E.

REVIEWER COMMENTS

Reviewer #1 (Remarks to the Author):

Review of the article " A FAN1 point mutation associated with accelerated Huntington's disease 2 progression alters its PCNA-mediated assembly on DNA " submitted by Aretz J et al
The manuscript by Aretz et al. makes a significant contribution to our understanding of FAN1 and PCNA's roles in Huntington's disease (HD), focusing on the R507H mutation associated with early-onset HD. The authors present a 2.6 Å X-ray structure of the FAN1 R507 mutant bound to DNA, alongside three cryo-EM structures of FAN1-DNA-PCNA complexes, including a ternary complex at 3.3 Å resolution. These structural insights illuminate the interactions within the ternary complex, revealing a direct interaction between R507 in FAN1 and D232 in PCNA. Notably, the DNA exhibits a 68° kink at the FAN1 active site.

Through SPR and MST analyses, the authors demonstrate that PCNA has weak intrinsic affinity for DNA, which is enhanced in the presence of FAN1. They also show that truncated FAN1 and mutations in R507 or D232 significantly reduce this affinity. Additionally, experiments using DNA with varying lengths of CAG repeats (1 to 4) showed no substantial differences in binding behavior.

The manuscript is well-written, with high-quality figures, and provides critical insights into DNA repair mechanisms and Huntington's disease pathology. This study represents an important advancement in the field and will appeal to a broad audience in molecular biology and neurodegenerative disease research.

We thank Reviewer 1 for the encouraging comments.

Reviewer #2 (Remarks to the Author):

Thanks to Reviewer 2 for identifying discrepancies.

Reviewer #3 (Remarks to the Author):

FAN1 possesses both 5' flap endonuclease and 5'-3' exonuclease activities and is involved in repairing DNA inter-strand crosslink damage. The structure of FAN1 and how FAN1 recognizes the DNA 5' flap or a branched DNA (with both 3' and 5' flaps) were determined over 10 years ago (PMID: 25500724; PMID: 25430771; PMID: 25547363), along with the discovery of the "n + 3" exonuclease mechanism (PMID: 25430771). The current manuscript attempts to address the recent observation that FAN1 mutation R507H is associated with the CAG expansion in Huntington's disease patients, which compromises the PCNA-activated FAN1 activity on these CAG expansions. The authors report the structures of the WT and R507H mutant FAN1 bound to PCNA and a DNA substrate, leading to the discovery that the single mutation abolishes a salt

bridge between FAN1 R507 and PCNA D232, thus weakening the interaction and explaining the compromised PCNA-stimulated FAN1 activity. They further performed in vitro binding and activity assays on DNA substrates containing 1 to 4 CAG repeat expansion, with overall anticipated results. A major weakness of the work is the use of flapped DNA substrate rather than a (CAG)_n bulge in their structural study, because FAN1 function on flapped DNA substrate is well understood thanks to previous structural and biochemical studies, and also because the DNA substrate used in structural work is different from biochemical assays which used CAG expansion DNA substrate, which is their current goal. Therefore, it remains unclear how FAN1 recognize and remove the CAG expansion. Overall, this work represents a modest incremental progress in this field, and one wonders if Communications Biology is a better fit.

We thank Reviewer 3 for the detailed comments and suggestions for edits. We do not believe that our work is just “modest incremental progress.” We fully acknowledge that previous studies made key structural discoveries of FAN1’s interaction with DNA. We not only provide cryo-EM structures of the FAN1-PCNA-DNA complex that have not been reported previously, we extend the structural work to include the disease-relevant FAN1 R507H variant. These findings allowed us to understand the mechanistic basis by which this FAN1 variant modulates the CAG repeat expansion in the huntingtin gene.

We appreciate this important point about using a DNA substrate with a CAG loop and have performed additional experiments to address it. Specifically, we resolved the cryo-EM structure of the FAN1-PCNA complex bound to a (CAG)_n loop DNA substrate at 3.6 Å resolution (Extended Data Figure 6A-C). This structure reveals that the FAN1-PCNA interface remains unchanged when compared to the previously resolved structure of FAN1-PCNA bound to a 5’ flap DNA substrate (PDB: 9EO1). The critical interaction between FAN1 R507 and PCNA D232 is preserved in both structures, highlighting the robustness of the FAN1-PCNA interaction regardless of DNA substrate type.

Furthermore, our findings are consistent with recent work by Li et al. (2024), who resolved cryo-EM structures of FAN1-PCNA bound to (CAG)_n loop DNA (PDB: 9CG4, 9CL7). Superimposition of our FAN1-PCNA-(CAG)_n structure with our FAN1-PCNA-5’ flap DNA structure (PDB: 9EO1) confirms that the overall architecture of the FAN1-PCNA interface remains unchanged, including the angle of the DNA upon FAN1 binding (Extended Data Figure 6D).

These structural results, together with biochemical assays using CAG repeat expansions, demonstrate that the structural insights derived from 5’ flap DNA studies are relevant and informative to FAN1’s activity on CAG repeat expansions. This effectively bridges the gap between structural and biochemical studies, addressing the reviewer’s concern.

We believe the combined use of high-resolution structures with model substrates, along with supporting biochemical and low-resolution data, provides a strong foundation for the conclusions drawn in this manuscript.

Major concerns:

1. Previous key discoveries on the structure and mechanism of FAN1 (PMID: 25500724; PMID: 25430771; PMID: 25547363) were not prominently cited and summarized in Introduction. Only one of them is cited in Introduction for mutation effect (line 53); another one merely mentioned in Results for the selection of DNA substrates (line 74). It is rather disappointing, and such practice is not acceptable.

We have modified the introduction to summarize the previous discoveries and indicate what we believe we are contributing to the field, filling in knowledge gaps that existed. We have explicitly summarized and cited the key structural studies on FAN1, including PMID: 25500724, PMID: 25430771, and PMID: 25547363. This ensures proper acknowledgment of foundational work and provides necessary context for our study. Please see lines 40-83 of the revised manuscript.

2. As mentioned above in summary, it is a major weakness of the work to use different DNA substrates for their structural and activity studies. The 5' flapped DNA substrate used in this work was already used in previous structural work. Therefore, the key question of how FAN1 recognizes CAG repeat expansion remains unclear.

Please see our response above, we have performed this analysis and the results are reported in Extended Data Figures 6A-D (legend lines 963-974). Please see line 135-139 in the manuscript for the description of the results. We also included a paragraph in the discussion section, lines 300-312.

2. Line 131 states in the presence of DNA, WT FAN1 and WT PCNA showed the highest binding affinity (Fig. 3C-D); but line 218 states "In line with our previous results, the PCNA-mediated activation of FAN1 WT was higher compared to FAN1 with the N-terminal truncation, FAN1 R507H and PCNA D232A (Fig. 5G)". And in Fig. 5C-D, with the presence of WT PCNA, the FAN1 R507, trFAN1 and trFAN1 R507 mutants had lower nuclease activity than without, and the first mutant had almost no activity. These observations are inconsistent.

We re-worded (lines 219-224): For PCNA D232A, trFAN1 WT ($k_{cat} = 5.5 \pm 2.4 \text{ s}^{-1}$) and R507H ($k_{cat} = 9.5 \pm 2.5 \text{ s}^{-1}$) activity remains unchanged while FAN1 WT and R507H activity could not and only partially ($k_{cat} = 2.0 \pm 0.7 \text{ s}^{-1}$) be restored, respectively (**Figure 5C-E; Extended Data Figure 8M**). Hence, the interaction with PCNA inhibited FAN1 activity on 5' flap DNA substrates, which could be restored by truncating the FAN1 N-terminus and introducing the FAN1 R507H and PCNA D232A mutations.

In addition, we removed "in line with our previous results" to avoid this misunderstanding and changed the sentence into: "In contrast to the 5'pG1/3T1 DNA substrate PCNA-mediated activation of FAN1 WT was observed on dsDNA which was higher compared to FAN1 with the N-terminal truncation, FAN1 R507H and PCNA D232A (**Figure 5G**)" (Lines 240-241)

Finally, we added "Taken together, our results demonstrate that FAN1 and PCNA form a ternary complex with DNA, that is required for PCNA-mediated FAN1 affinity modulation on different DNA substrates." to highlight the difference in affinity modulation. (Lines 250-251)

3. Line 240, ... the previously unresolved loop between residues 507-518: Untrue, as this region was resolved in one of the crystallographic works mentioned above (PDB 4RI8 chain A).

We have revised the manuscript to acknowledge that the loop between residues 507-518 was resolved in PDB 4RI8 (Wang et al., 2014). The revised statement now correctly distinguishes between prior crystallographic findings and our new structural observations. (Lines 263-269).

Minor issue: Missing positive and negative controls in FAN1 endonuclease activity assays, i.e., with or without WT FAN1 in Fig. 5C-E.

Typos:

Line 15, two “Huntington’s disease”.

Deleted

Line 26, add “_” to FAN1 R507H in “FAN1 R507H-PCNA-DNA”, also replace the hyphen (-) with dash (–) as in line 22 for readability, also check other places, e.g., line 97.

Dashes replaced with hyphens

Line 42-45, provide a domain architecture diagram of FAN1 can be helpful.

Provided as Supplementary Figure f

Line 71, use the prime symbol (5′) rather than apostrophe (') throughout the text.

Replaced

Line 83, Fig. 1B was cited before 1A.

For the figure, we think the ternary complex as 1A is more aesthetically pleasing in the reverse order.

Line 92, ... studies6..., cite more or use singular form.

Completed

Line 113, “ICDL” should be “IDCL”.

Fixed

Line 136, ... complex between FAN1, PCNA and DNA requires at least two interactions with the FAN1 N-terminus... the present data don’t lead to such conclusion.

As both the truncation of the N-terminus as well as the introduction of the R507H mutation lead to a reduced PCNA affinity, two interaction sites have to exist, which are located in the flexible loop between 507-518 and in the N-terminus of FAN1.

Line 201, “trFAN1 WT” is confusing, same for the figure legends e.g., Extended Data Fig. 8B. Also the number of kcat is inconsistent with Fig. 3D.

Corrected, thanks for pointing this out.

Line 416 and 729 these citations refer to the same paper.

Removed the 2nd citation.

Line 822, Fig. 1 should be Fig. 3.

Thank you, fixed.

Line 1016, missing concentration in Fig. 3B. The numbers above the histogram in Fig. 3D are confusing.

Concentration added. Numbers above 3D are p-values as requested by the journal.

Line 1033, 1035, extra spaces within “acquired” and Local “Motion Correction” in Extended Data Figs. 3 and 4 legends.

We double checked that there is indeed a space.